# Development of an LC-MS/MS Method for Quantification of Sapitinib in Human Liver Microsomes: In Silico and In Vitro Metabolic Stability Evaluation

**DOI:** 10.3390/molecules28052322

**Published:** 2023-03-02

**Authors:** Mohamed W. Attwa, Haitham AlRabiah, Gamal A. E. Mostafa, Adnan A. Kadi

**Affiliations:** 1Department of Pharmaceutical Chemistry, College of Pharmacy, King Saud University, Riyadh 11451, Saudi Arabia; 2Students’ University Hospital, Mansoura University, Mansoura 35516, Egypt; 3Micro-Analytical Laboratory, Applied Organic Chemistry Department, National Research Center, Cairo 12622, Egypt

**Keywords:** sapitinib, metabolic stability, in vitro half-life, intrinsic clearance, LC-MS/MS, WhichP450 module

## Abstract

Sapitinib (AZD8931, SPT) is a tyrosine kinase inhibitor of the epidermal growth factor receptor (EGFR) family (pan-erbB). In multiple tumor cell lines, STP has been shown to be a much more potent inhibitor of EGF-driven cellular proliferation than gefitinib. In the current study, a highly sensitive, rapid, and specific LC-MS/MS analytical method for the estimation of SPT in human liver microsomes (HLMs) was established with application to metabolic stability assessment. The LC-MS/MS analytical method was validated in terms of linearity, selectivity, precision, accuracy, matrix effect, extraction recovery, carryover, and stability following the FDA guidelines for bioanalytical method validation. SPT was detected using electrospray ionization (ESI) as an ionization source under multiple reaction monitoring (MRM) in the positive ion mode. The IS-normalized matrix factor and extraction recovery were acceptable for the bioanalysis of SPT. The SPT calibration curve was linear, from 1 ng/mL to 3000 ng/mL HLM matrix samples, with a linear regression equation of y = 1.7298x + 3.62941 (r^2^ = 0.9949). The intraday and interday accuracy and precision values of the LC-MS/MS method were −1.45–7.25% and 0.29–6.31%, respectively. SPT and filgotinib (FGT) (internal standard; IS) were separated through the use of an isocratic mobile phase system with a Luna 3 µm PFP(2) column (150 × 4.6 mm) stationary phase column. The limit of quantification (LOQ) was 0.88 ng/mL, confirming the LC-MS/MS method sensitivity. The intrinsic clearance and in vitro half-life of STP were 38.48 mL/min/kg and 21.07 min, respectively. STP exhibited a moderate extraction ratio that revealed good bioavailability. The literature review demonstrated that the current analytical method is the first developed LC-MS/MS method for the quantification of SPT in an HLM matrix with application to SPT metabolic stability evaluation.

## 1. Introduction

Cancer is considered the main cause of death worldwide and is known as the uncontrolled division of cells in a certain part of the body that can also spread to different parts of the body. It occurs due to the damage of genes that are responsible for controlling different cellular functions that can allow a cell to become malignant. [1]. Molecular-targeting strategies have been used for cancer management based on the suppressor genes and tumor oncogenes participating in human cancer development [2]. The epidermal growth factor receptor (EGFR) family, also called the erbB protein family, plays a major role in tumor cell proliferation and tumor vascularization [3].

Sapitinib (SPT) is a tyrosine kinase inhibitor (TKI) of the EGFR family (Figure 1). SPT is reported to be a stronger inhibitor of EGF-driven cellular propagation in different tumor cell lines compared to gefitinib, the first-line clinical TKI. SPT binds to and inhibits erbB tyrosine receptor kinases, which may lead to the inhibition of cellular angiogenesis and propagation of tumors expressing erbB. SPT exclusively gives similar inhibition of EGFR, erbB2, and erbB3 signaling and shows more antitumor activity in certain preclinical models compared to other, narrower-spectrum agents with erbB receptor inhibition [4].

The metabolic stability of a drug is the lability to metabolism and is calculated as intrinsic clearance (CL_int_) and in vitro half-life (t_1/2_). CL_int_ is defined as the liver’s capability to metabolize a drug in the blood. Half-life (t_1/2_) is defined as the time required for a 50% decrease in parent drug concentration due to metabolism. In vitro Cl_int_ and t_1/2_ in HLMs have been calculated using the well-stirred model (in vitro t_1/2_ approach) [5,6], as it is an often utilized model in drug metabolism experiments owing to its simplicity. These parameters (in vitro t_1/2_ and Cl_int_) can be used for calculating different physiological parameters (e.g., in vivo t_1/2_ and liver clearance). Drug bioavailability estimation provides a good approach to proposing in vivo metabolic reactions. If an examined drug shows a fast metabolic rate, it exhibits low in vivo bioavailability value and a short duration of action [7,8,9,10].

SPT is mainly metabolized in the liver by major metabolic pathways by oxidation and amine or ether cleavage around the piperidine ring with following sulphate or glucuronide conjugation [11,12]. SPT treatment is associated with four adverse effects, i.e., catheter-site-related reactions, oropharyngeal pain, blepharitis, and gingival pain [4]. Therefore, metabolic stability (in vitro and in silico) is necessary for developing a new drug with a better metabolic stability profile [13,14]. Drugs with high metabolic rates are expected to show short durations of action and low in vivo bioavailability [15].

A literature review revealed that there is no reported article for a fully validated LC-MS/MS method for SPT quantification in an HLM matrix. There are two articles published for the qualitative identification of metabolites of SPT [11,12]. Peter et al. reported LC-MS/MS for the quantitative estimation of SPT and *O*-desmethyl metabolites without providing any details about the analytical chromatographic parameters, and no method validation was performed [12]. Attwa et al. reported a qualitative analytical method for the profiling of metabolites and reactive intermediates of SPT without determining the rate of metabolism with time [11]. In addition, high concentrations of SPT (10 µM/mL) are used in metabolic profiling studies, which are not performed for metabolic stability studies of SPT (1 µM/mL) and need to be lower than the Michaelis–Menten constant to construct a linear relationship between STP metabolic rate and the time of metabolic incubation.

To determine SPT metabolic stability in vitro, this study aims to validate and develop a reliable LC-MS/MS method. Protein precipitation is applied for drug extraction from an HLM matrix. All analytical parameters, for example, calibration, recovery, accuracy, and precision, are determined according to FDA guidelines. SPT is tested for its metabolism lability in HLMs (liver) using in silico software (StarDrop WhichP450 model) before the initiation of the practical experiment to confirm the value of the current work and to save time and resources [16]. Understanding how SPT is metabolized and its metabolic stability allows the identification and analysis of its proposed metabolites and guides the design of molecules with better metabolic behavior. The developed LC-MS/MS method is applied for a practical evaluation of the intrinsic clearance (Cl_int_) and in vitro half-life (t_1/2_) of SPT [13]. In silico StarDrop WhichP450 model software and in vitro LC-MS/MS experiments are used for the assessment of SPT metabolic stability to give more information about the metabolic rate of SPT and to allow in vivo bioavailability assessment.

## 2. Results and Discussion

### 2.1. In Silico SPT Metabolic Stability

The WhichP450 model predicted the major metabolizing isoform (CYP3A4) for SPT metabolism, as shown in the pie chart (Figure 2A). The regioselectivity map indicates the predicted sites of metabolism for SPT (Figure 2B). The metabolic landscape (Figure 2C) predicted the SPT metabolic lability of the active sites to improve the understanding of the SPT metabolic rate [17,18,19]. The CSL (0.9947) revealed high SPT metabolic lability; thus, the established method was applied for SPT metabolic stability estimation (Figure 2). The IUPAC name of SPT is 2-[4-[4-(3-chloro-2-fluoroanilino)-7-methoxyquinazolin-6-yl]oxypiperidin-1-yl]-*N*-methylacetamide. The outcomes showed that C1, C5 of the *N*-methylacetamide group, and C29 of 3-chloro-2-fluoroanilino were labile to metabolism, while C20 of the methoxy group was moderately labile to metabolism. The in silico results showed that the *N*-methylacetamide group (C1 and C5) was the chief cause of SPT metabolic lability, as shown by the CSL exhibited in Figure 2 (value of 0.9947 revealing high lability to metabolism), which matched with the in vitro experiments (discussed later).

### 2.2. LC-MS/MS Method Development

The reversed-phase C18 column was tested for separation in the current work (quantitative work); analytes were retained but generated chromatographic peak tailing, poor separation, and longer retention time. Therefore, the best results were attained using the Luna 3 µm PFP(2) column (150 × 4.6 mm) regarding peak shape and retention time. Both analytes (STP and FGT) were separated in a short running time (3 min) after injecting 5 µL into the UPLC-TQD MS system. In the established method, an isocratic mobile phase in 3 min of run time using a smaller flow rate (0.4 mL) was used. Moreover, the calibration curve was linear in the range of 1 to 3000 ng/mL. Different experiments were performed to select the most optimum condition for the separation, extraction, and analysis of SPT and FGT chromatographic peaks in perfect shape and in a short running time, as mentioned in Table 1.

The MRM mass analyzer mode was used for quantifying STP and FGT in order to avoid interference from constituents of the HLM matrix that elevated the sensitivity of the developed analytical method. The predicted fragmentation patterns were explained (Figure 3).

FGT was chosen as the IS in the developed LC-MS/MS analytical method for estimating SPT in the HLM matrix because of three reasons. First, both analytes (SPT and FGT) were extracted from the HLM matrix utilizing the protein precipitation extraction method with an efficient percentage of recovery for SPT (101.33 ± 2.11) and FGT (106.12 ± 4.92%). Second, the eluted peaks of FGT (1.24 min) and SPT (2.17 min) were detected in 3 min with a reasonable separation that was considered a fast analytical method saving time and organic solvent (green chemistry). Third, there was no prescription approved for the concurrent use of both anticancer drugs (SPT and FGT) for the same patient. Therefore, the developed analytical method could be used for therapeutic drug monitoring or pharmacokinetics studies of SPT. No observed carryover was found for SPT in the HLM negative control MRM chromatograms (Figure 4A) or the positive control MRM chromatograms (Figure 4C). Figure 4B illustrates the overlaid MRM chromatograms of SLP calibration levels (1 ng/mL to 3000 ng/mL).

### 2.3. Validation of UPLC-TQD MS Method

#### 2.3.1. Specificity

The specificity of the established method was approved by the good separation of the chromatographic peaks of STP and FGT, as shown in Figure 4. In addition, there were no interference with the chromatographic peaks of STP and FGT from the HLM matrix constituents (Figure 4A). No carryover effect of STP was seen in the control blank MRM chromatograms (Figure 4C).

#### 2.3.2. Sensitivity and Linearity

The linearity of the established method was confirmed statistically in the range of 1 to 3000 ng/mL with a regression equation of y = 6.561x + 8.415 and a coefficient of variation (R^2^: 0.9993) by injecting eight SPT calibration standards and quality controls in the HLM matrix and then back-calculating as unknowns. Weighting (1/x) of the calibration line was applied. The RSD values for the six repeats (eleven calibration standards and QCs) were <3.72% (Table 2). The LOD and LOQ were 0.27 ng/mL and 0.82 ng/mL, respectively.

#### 2.3.3. Precision and Accuracy

The interday and intraday precision and accuracy of the established method were confirmed by injecting six repeats (four QCs) over three following days and twelve repeats (four QCs) on the same day, respectively. The outcomes were in the acceptable range according to the FDA guidelines. The intraday and interday accuracy and precision values of the established method were −1.45–7.25% and 0.29–6.31%, respectively (Table 3).

#### 2.3.4. SPT Extraction Recovery and Matrix Effects of HLMs

The efficiency of the selected method of analyte (STP and FGT) extraction (protein precipitation using ACN) was approved by injecting six repeats (four QCs) in the HLM matrix and comparing it with QCs prepared in mobile phase. The results revealed high extraction recovery rates for STP (101.33 ± 2.11 and RSD < 4.1%) and FGT (102.32 ± 3.62%.). The HLM matrix had no effect on the ionization degree of parent ion (STP or FGT) ionization, and that was revealed by injecting two sets of samples of HLM matrix. Sample set 1 was spiked with STP LQC (3 ng/mL) and FGT (1000 ng/mL), while sample set 2 was created using the mobile phase instead of the HLM matrix. The HLMs containing STP and FGT showed a matrix effect of 101.74 ± 4.15% and 102.24 ± 2.26%, respectively. The IS-normalized ME was 0.995, which was accepted following FDA guidelines. Therefore, these outcomes revealed that the HLM matrix showed no observed effect on the degree of parent ionization of either FGT or STP.

#### 2.3.5. Stability of SPT in Stock Solution and HLM Matrix

SPT stabilities in stock solution (DMSO) and in the HLM matrix were evaluated. SPT showed perfect stability in DMSO after storage for 28 days at −80 °C. The stability values ranged from 98.8 to 101.3% for SPT in HLM matrix samples (Table 4). There was no observed loss of SPT after autosampler storage, short-term storage, three freeze–thaw cycles, and long-term storage. The results reveal that good stability of SPT was achieved.

### 2.4. In Vitro Metabolic Stability of STP

In in vitro metabolic stability assessment work, the analyte (STP) concentration should be 1 µM/mL in the HLM incubation mixture to be lower than the Michaelis–Menten constant and construct a linear relationship between STP metabolic rate and the time of metabolic incubation. In addition, HLM microsomal protein should be 1 mg/mL in the HLM incubation mixture to escape protein binding. The first STP metabolic curve was established by plotting the time points (x-axis) from 0 to 70 min against the percentage of STP concentration remaining compared to the zero-time concentration (y-axis) (Figure 5A). The linear part (0–30 min) of the constructed curve was chosen to construct another natural logarithmic curve of incubation time points (0 to 30 min) against the natural logarithm (ln) of the percentage of STP remaining (Figure 5B). The slope of the second curve (0.0329) was considered the STP metabolic rate constant, while its linear regression equation (y = −0.0329x + 4.6097 with R² = 0.998) was used for calculating STP in vitro t_1/2_ (Table 5). From the previous regression equation, the slope was 0.0329 and was used to calculate the in vitro t_1/2_, where in vitro t_1/2_ = ln2/slope, so the in vitro t_1/2_ was 21.07 min. STP intrinsic clearance was calculated according to the in vitro t_1/2_ method, so the Cl_int_ of STP was 38.48 mL/min/kg Metabolic stability, which provides an estimate of the lability to biotransformation, could be calculated from the HLM incubations (in vitro t_1/2_ and Cl_int_.) According to the scoring provided in McNaney et al. [20], STP is considered an intermediate clearance drug, which shows the importance of this type of kinetic experiment for more accurate metabolic lability predictions. By using other software (Cloe PK software and simulations), these outcomes could also be utilized to propose STP in vivo pharmacokinetics [21].

## 3. Material and Methods

### 3.1. Materials and Instruments

All solvents used in the current work were of high-performance liquid chromatographic (HPLC) grade. All reference powders and solid chemicals were of analytical (AR) grade. The two analytes, filgotinib (Synonyms: GLPG0634; Cat. No.: HY-18300; purity: 99.37%) and SPT (Synonyms: AZD-8931; Cat. No.: HY-13050; purity: 99.93%) were procured from MedChem Express (Princeton, NJ, USA). Formic acid, ammonium formate, acetonitrile (ACN) and HLMs (20 mg/mL) were purchased from Sigma-Aldrich (St. Louis, MO, USA). After delivery of the HLMs, they were stored at −70 °C until use. An in-house Milli-Q plus purification water system (Millipore; Billerica, MA, USA) was used for generation of water at the HPLC-grade level. An ultra-performance liquid chromatography triple quadrupole mass spectrometry (UPLC-TQD MS) instrument composed of an Acquity TQD MS (model code (TQD) and serial number (QBB1203)) and an Acquity UPLC (model code (UPH) and serial number (H10UPH)) was used for mass detection and quantification of analyte (TQD MS) peaks after being chromatographically separated from the HLM matrix components. MassLynx 4.1 software (Version 4.1, SCN 805) controlled the UPLC-TQD MS system. Interpretation and processing of the collected data were performed using the QuanLynx application manager included with the MassLynx 4.1 software package. The mass spectrometric parameters of both analytes (SPT and FGT) were tuned using a smart tool of IntelliStart^®^ software included in the operating software. A vacuum pump (Sogevac^®^; Murrysville, PA, USA) was used for generating the necessary vacuum inside the mass analyzer. A nitrogen generator (Peak Scientific; Renfrewshire, Scotland, UK) was used for the generation of nitrogen as a drying gas inside the electrospray ionization source for the drying of mobile phase droplets. Argon gas (local supplier; 99.999%) cylinders were used to supply argon as a collision gas in the second quadrupole (collision cell) of the TQD mass analyzer.

### 3.2. In Silico Evaluation of SPT Metabolic Stability

WhichP450 StarDrop software (Optibrium Ltd.; Cambridge, MA, USA) proposed the metabolism regioselectivity of the seven major drug-metabolizing isoforms of Cytochrome P450 enzymes (CYP3A4, CYP2D5, CYP1A2, CYP2C9, CYP2C8, CYP2C19, and CYP2E1). The results are represented as composite site lability (CSL) values indicating the target analyte (SPT) metabolic lability. CSL was considered an important parameter in predicting the metabolic rate of SPT before running the in vitro experiments to confirm the importance of the current work in order to save time and resources. The SPT (CNC(=O)CN1CCC(CC1)OC2=C(C=C3C(=C2)C(=NC=N3)NC4=C(C(=CC=C4)Cl)F)OC) SMILES format was attached to StarDrop software for CSL prediction. To assess the metabolic lability of SPT, the individual atom labilities were gathered to compute the CSL, revealing the overall metabolic stability for SPT [22,23,24] according to Equation (1):(1)CSL =ktotalktotal+kw
where *k_w_* is the rate constant for water formation.

### 3.3. UPLC-TQD MS Optimized Parameters

#### 3.3.1. UPLC Parameters

UPLC parameters participating in the chromatographic separation of analytes (SPT and FGT), such as mobile phase composition, stationary phase nature, and pH of the mobile phase, were optimized to obtain the best separation and the maximum intensity of analyte peaks. First, the mobile phase was composed of an aqueous part and an organic part. The aqueous part (0.1% 10 mM ammonium formate in H_2_O) represented 45% of the mobile phase, and the pH was adjusted (using pH meter) to 4.8 using a few drops formic acid. Increasing the pH value to more than 4.8 caused chromatographic peak tailing and long elution time. The organic part (ACN) represented 55% of the mobile phase. Increasing the percentage of ACN (more than 55%) generated overlapped and poorly separated chromatographic peaks, while decreasing the percentage of ACN (less than 55%) generated long elution time. Second, the selected stationary phase was a Luna 3 µm PFP(2) column (150 × 4.6 mm). PFP(2) is a new pentafluorophenyl with a propyl linkage stationary phase that gives different retention methods that are unique compared to other reversed stationary phases.

#### 3.3.2. TQD MS Parameters

TQD MS parameters participating in the mass spectrometric analysis and detection of the two analytes (SPT and FGT) were optimized to obtain the maximum ionization and sensitivity of eluted chromatographic peaks from the UPLC system after entering through an ESI source that was operated in the positive due to the basic nature of the two analytes (contain basic nirogens) that could capture protons from the mobile phase. Tuning for the mass spectrometric parameters of SPT (molecular formula: C_23_H_25_ClFN_5_O_3_) and FGT (molecular formula: C_21_H_23_N_5_O_3_S) was performed using IntelliStart^®^ software in the combined mode (fluidics and LC) by direct infusion into the mobile phase from 1 µg/mL of the stock solution of the two analytes (one at a time). The flow rate of the cone gas was adjusted to 100 L/H. Nitrogen gas (650 L/h) from a nitrogen generator was used as a drying gas inside the ESI source at 350 °C. The multiple-reaction-monitoring (MRM) mass analyzer mode (parent ion to two daughter ions) was used for quantifying the two analytes (SPT and FGT) to elevate the sensitivity and selectivity of the established analytical method. Argon (0.14 mL/min) gas from an argon cylinder was used as a collision gas for breaking down the parent ion into daughter ions inside the collision cell. The capillary voltage, extractor voltage, and RF lens were set at 4 (kV), 3.0 (V), and 0.1 (V), respectively. The cone voltages for SPT and FGT were adjusted to 42 (V) and 38 (V), respectively. The dwell time for SPT and FGT mass transitions was 0.025 s. All MRM parameters and mass transitions of SPT and FGT (IS) are listed in Table 6.

### 3.4. Preparation of SPT Working Solutions

Both analytes (STP and FGT) showed good solubility in dimethyl sulfoxide organic solvent (DMSO) at ≥ 33 mg/mL (69.63 mM) and 25 mg/mL (58.75 mM; need ultrasonic), respectively. Therefore, stock solutions (1 mg/mL) of STP and FGT were prepared in DMSO due to the solubility and stability of the two analytes in DMSO. Sequential dilutions of the SPT stock solution (1 mg/mL) with mobile phase were performed to prepare working solutions (WK) at 100 µg/mL (STP WK1), 10 µg/mL (STP WK2), and 1 µg/mL (STP WK3). Multiple dilutions of the FGT stock solution (1 mg/mL) were performed to prepare FGT WK3 at 10 µg/mL.

### 3.5. Construction of Calibration Curve of SPT

HLMs were deactivated before spiking with the two analytes (SPT and FGT) to avoid the effect of metabolic enzymes on the concentrations of the analytes during the validation steps. Deactivation of the HLM matrix was performed using DMSO, as it quenched metabolic reactions at 2% concentration [25] with slight warming at 50 °C for 5 min since heat stops HLM activity [26,27]. The HLM matrix was prepared at a concentration of 1 mg microsomal protein/mL by diluting 30 µL deactivated HLMs to 1 mL with incubation buffer (0.1 M sodium phosphate buffer: pH 7.4) containing 1 mM NADPH to simulate in vitro experiments of metabolic stability evaluation. STP calibration standards were prepared by the multistep dilution of STP WK2 and STP WK3 with the deactivated HLM matrix, generating eight levels of 1, 15, 50, 150, 300, 500, 1500, and 3000 ng/mL and keeping the HLM matrix volume no less than 90% of the prepared solution in order to decrease the influence of dilution compared to real samples. These STP calibration standards were used for making a calibration curve. Four STP levels were selected as quality controls (QCs) for the STP validation procedure: LLQC (1 ng/mL), LQC (3 ng/mL), MQC (900 ng/mL), and HQC (2400 ng/mL). QCs were used as unknowns, and the concentrations were determined using a freshly prepared STP calibration curve. One hundred microliters of FGT WK1 was added as the IS (1000 ng/mL) to 1 mL of calibration levels and QCs.

### 3.6. Extraction of SPT and FGT from HLM Matrix

Protein precipitation using ACN was used for extraction of the two analytes (STP and FGT) from the HLM matrix following the next steps. First, two mL of ACN was added to 1 mL of the STP calibration levels, quality controls, or unknown samples. Second, after shaking for 5 min, all samples were centrifuged at 14,000 rpm for 12 min (thermostated centrifuge at 4 °C) for purifying the supernatants. Third, filtration of 1 mL of the supernatant through a syringe filter (0.22 µm) was performed in HPLC vials to confirm the suitability of samples for injection into the UPLC-TQD MS system. A positive control (HLM matrix plus IS) and a negative control (HLM matrix) were prepared using the same three steps previously mentioned. These controls were used to confirm the absence of interference from HLM matrix constituents at the elution times of both analytes (STP and FGT). A STP linear calibration curve was constructed by plotting the peak area ratio of STP to FGT (y-axis) versus STP nominal values (x-axis). The linear regression equation (y = ax + b; r^2^) was used for revealing the linearity range of the constructed calibration curve.

### 3.7. Validation of UPLC-TQD MS Method

The bioanalytical validation steps of the UPLC-TQD MS method were performed using linearity, sensitivity, specificity, accuracy, precision, matrix effect, extraction recovery, and stability according to USFDA general regulations [28].

#### 3.7.1. Specificity

The specificity of the UPLC-TQD MS analytical method was evaluated by analyzing six blank HLM matrix batches after performing the protein precipitation extraction steps. Then, these extracts (5 µL) were injected into the UPLC-TQD MS and tested for any interference peaks at STP or FGT elution times, as well as comparing the MRM chromatogram of spiked HLM matrix samples with the two analytes (STP and FGT). MRM mode was used to reduce the carryover effects of the two analytes (STP and FGT) in the mass analyzer, as approved by injecting the negative control HLMs without any analytes.

#### 3.7.2. Linearity and Sensitivity

The sensitivity and linearity of the established UPLC-TQD MS method were evaluated by injecting freshly prepared calibration curves (eight standards) and quality controls of STP in an HLM matrix in the same day and then back-calculating all data unknowns using the linear regression equation of each curve. The LOD and LOQ were computed as stated in the Pharmacopeia with the slope of the developed calibration curve and the standard deviation (SD) of the intercept using Equation (2) and Equation (3), respectively:(2)LOD =3.3×SD of the interceptSlope
(3)LOQ =10×SD of the interceptSlope

The linearity of the UPLC-TQD MS method was revealed by utilizing the least squares statistical method (y = ax + b) and the coefficient of variation (R^2^).

#### 3.7.3. Accuracy and Precision

The interday and intraday precision and accuracy of the established UPLC-TQD MS method were revealed by injecting 6 replicates of STP QCs over 3 following days and 12 replicates of STP QCs on the same day, respectively, according to USFDA validation guidelines. Accuracy and precision of the current UPLC-TQD MS method were expressed as the percentage error (RE %) and percentage relative standard deviation (RSD %), respectively, as calculated in Equation (4) and Equation (5), respectively.
(4) RSD % =SD Mean
(5)RE %=average concentration−proposed concentration proposed concentration×100

#### 3.7.4. Matrix Effect and Extraction Recovery

The extraction recovery of STP from the HLM matrix and the influence of the HLMs on the ionization of STP were evaluated by injecting four QC samples. The efficiency of the selected extraction (protein precipitation using ACN) method for the two analytes (STP and FGT) was confirmed by injecting 6 replicates of QCs in the HLM matrix (B) and comparing it with QCs that were prepared in mobile phase (A). The percentage recovery of the analytes was calculated as the ratio of B/A × 100. The absence of HLM matrix effect on the degree of ionization and the formation of parent ions of both analytes (STP and FGT) was verified by injecting two groups of samples. The HLM matrix (group 1) was spiked with STP LQC (3 ng/mL) and FGT (1000 ng/mL), while group 2 was created using the mobile phase instead of the HLM matrix. The matrix effects (MEs) for STP and FGT were calculated utilizing Equation (6). The IS-normalized ME was calculated utilizing Equation (7).
(6)Matrix effect of STP or FGT=mean peak area ratio Set 1Set 2×100
(7)IS normalized ME=matrix effect of STPmatrix effect of FGT IS

### 3.8. Stability

SPT stability in stock preparations and in HLM matrix samples were assessed using laboratory conditions that a material could be subjected to before analysis, including after short-term storage, autosampler storage, three freeze–thaw cycles, and long-term storage.

### 3.9. In Vitro Evaluation of SPT Metabolic Stability

The STP metabolic stability parameters (the CL_int_ and in vitro t_1/2_) were computed by the estimation of the percentage of remaining STP concentration after incubation with an HLM matrix (active) that contained NADPH as a cofactor, which started the incubation metabolic reactions. The metabolic procedures for the incubation of STP and HLM matrix samples were performed in four steps. The first step (conditioning) was pre-incubation of 1 µL of STP (1 mM) with active HLM matrix without the addition of a cofactor (NADPH) at 37 °C for 10 min in order to attain the optimum conditions for enzymatic metabolic reactions. The second step (incubation) of initiating the metabolic incubation was performed by adding 1 mM NADPH. The third step (quenching) of termination of the metabolic incubation was performed by adding 2 mL of ACN. The fourth step (IS addition) was the addition of 100 µL of FGT WK3 just before the addition of ACN in order to avoid the metabolic effect on the FGT concentration. Termination of the incubation was performed at selected time points: 0, 2.5, 7.5, 15, 20, 30, 40, 50, 60, and 70 min. The protein precipitation extraction steps were performed following the same steps as explained above [29].

The STP concentration in HLM incubation mixtures was computed using the linear regression equation of a freshly constructed STP calibration curve. First, an STP metabolic stability curve was constructed by plotting the quenched time points (x-axis) from 0 to 70 min against the percentage of STP concentration remaining compared to the zero-time concentration (100%) (y-axis). Then, points of the constructed curve that showed linearity were selected to establish a logarithmic curve by plotting the natural logarithm (ln) of the percentage of STP remaining for the selected points against incubation time points (0 to 30 min). The slope of the second established curve showed the rate constant for the STP metabolic stability and was utilized to calculate the in vitro t_1/2_, where the in vitro t_1/2_ = ln2/slope. Then, the STP CL_int_ (µL/min/mg) was calculated [20] using a value of 45 mg of HLM matrix (microsomal protein) per gram of liver tissue and 26 g for liver tissue per kilogram of body weight [30] (Equation (8)).
(8)$${\mathrm{CL}}_{\mathrm{int},}=\frac{0.693}{{\mathrm{in}\;\mathrm{vitro}\;t\;}_{\text{½}}}\times \frac{\mathrm{mL}\;\mathrm{incubation}}{\mathrm{mg}\;\mathrm{microsomes}}\times \frac{\mathrm{mg}\;\mathrm{microsomal}\;\mathrm{proteins}}{g\;\mathrm{liver}}\times \frac{g\;\mathrm{liver}}{\mathrm{Kg}\;b.\;w.}$$

## 4. Discussion

A fully validated LC-MS/MS analytical method was established for the estimation of SPT in an HLM matrix ranging from 1 ng/mL to 3000 ng/mL. The P450 StarDrop software results for CSL (0.9947) revealed high SPT metabolic lability that matched with the metabolic stability experiment (Cl_int_: 38.48 mL/min/kg), which was considered higher than previously studied tyrosine kinase inhibitors (e.g., tandutinib) [31,32,33,34]. The in silico results showed that the *N*-methylacetamide group (C1 and C5) was the main reason for SPT metabolic lability, which was previously proved in a metabolite identification study [11]. In our previous work about SPT metabolite identification, six metabolites were identified in the same metabolic spots that were predicted by P450 StarDrop software. The metabolic hydroxylation of C7 and C11 was seen in practical experiments (Figure 6).

From all the results of the previous qualitative work for SPT metabolite identification and the current work for the quantitative estimation of the rate of SPT metabolism, in silico software (StarDrop software) could be used in an efficient way to guide practical experiments to save time and resources, especially in the first steps of drug design.

## 5. Conclusions

A validated LC-MS/MS method was developed for quantifying STP in an HLM matrix and applied for assessment of its metabolic stability. It is considered the first validated LC-MS/MS method reported for STP quantification in an HLM matrix. The LC-MS/MS method showed good selectivity and sensitivity. The established method also showed high recovery of analytes (FGT and STP) from the HLM matrix using a protein precipitation extraction method. The consumption of less organic solvent (ACN) in the mobile phase made the analytical method ecofriendly. In vitro HLM metabolic incubations were performed to verify the outcomes of the in silico P450 StarDrop metabolic model. The results of the metabolic stability of STP (moderate CL_int_ of 38.48 mL/min/kg and in vitro t_1/2_ value of 21.07 min) showed that STP was a moderate clearance drug. Therefore, good in vivo bioavailability could be expected. From these results, we propose that STP could be given to patients without the effects of dose accumulation inside the human body or fast elimination by the liver. Future work may be conducted using in silico and in vitro tools for designing new drugs with increased metabolic stability. The in vitro incubation results of STP agreed the results of the in silico software, which confirmed the significance of in silico metabolic lability studies for saving effort and resources.

## Figures and Tables

**Figure 1 molecules-28-02322-f001:**
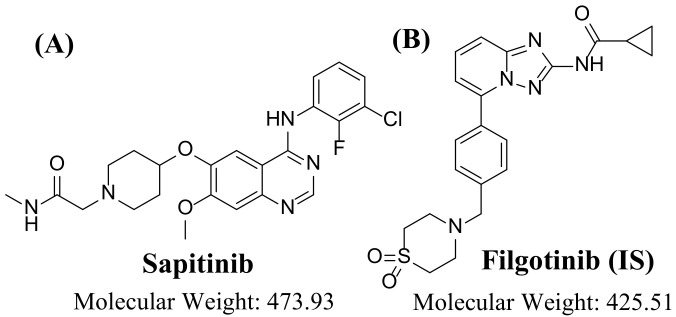
Chemical structures of (**A**) sapitinib and (**B**) filgotinib (internal standard; IS).

**Figure 2 molecules-28-02322-f002:**
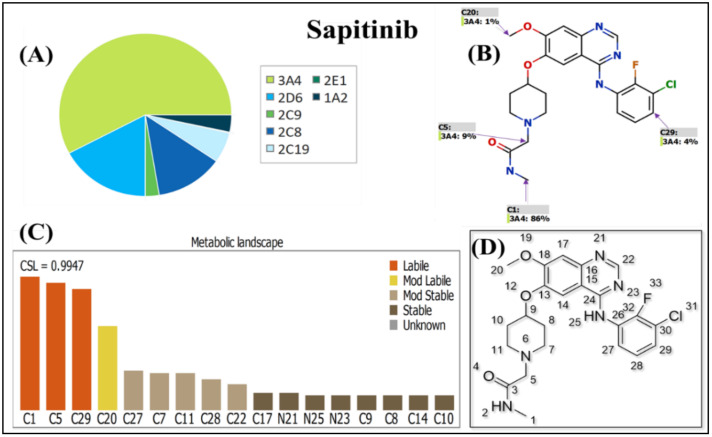
WhichP450 model prediction of major metabolizing isoform (CYP3A4) for SPT metabolism, as shown by pie chart (**A**). Regioselectivity map indicates predicted sites of metabolism for SPT (**B**). Metabolic landscape showing CSL of SPT (0.9797) revealed a high metabolic rate (**C**). Chemical structure of SPT with numbering system (**D**). These data were produced using StarDrop P450 metabolism model.

**Figure 3 molecules-28-02322-f003:**
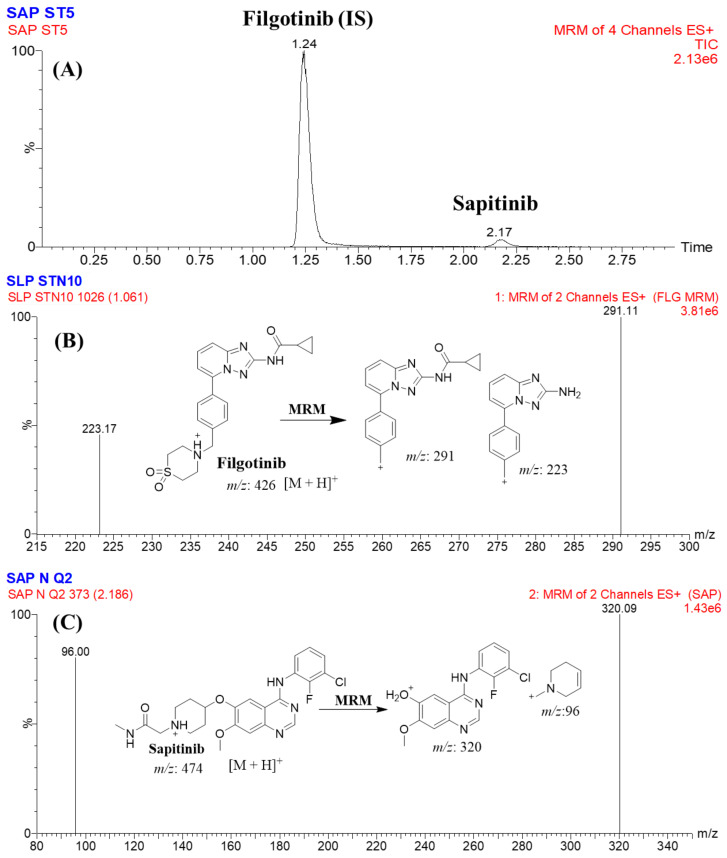
MRM chromatogram of the lower-quality control showing SPT peak (2.17 min) and FGT peak (1.24 min) (**A**). MRM mass spectrum of FGT (**B**). MRM mass spectrum of SPT (**C**).

**Figure 4 molecules-28-02322-f004:**
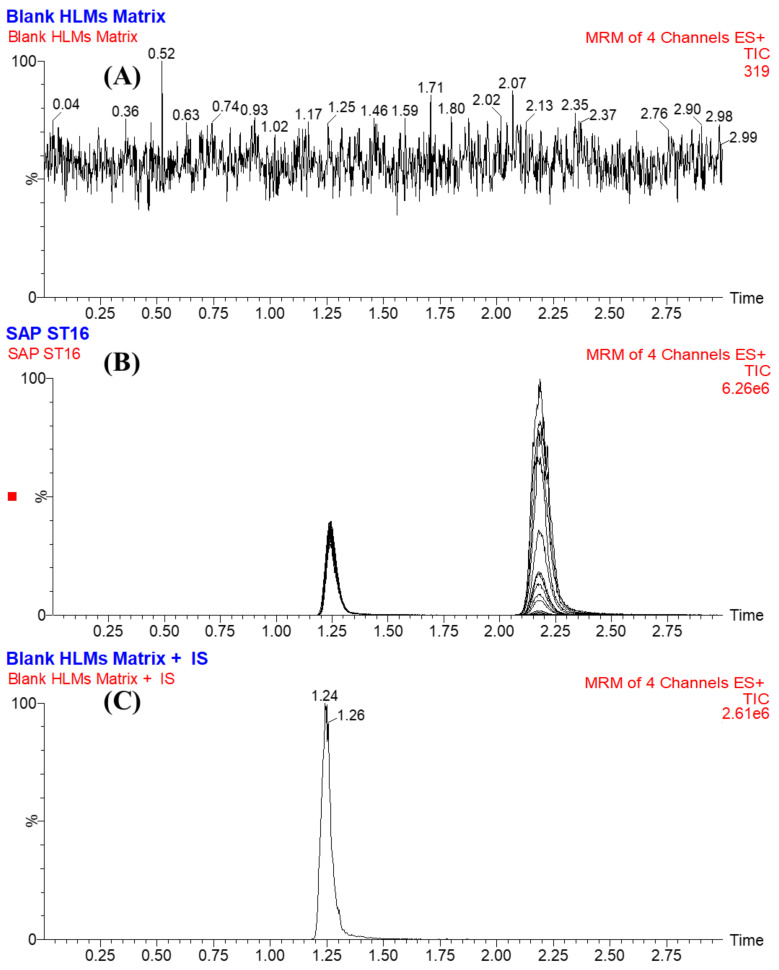
Blank HLM matrix showing no interfering peaks at the retention times of SPT and FGT (**A**) and overlaid MRM chromatograms of the SPT calibration levels (**B**) showing the SPT peak (2.1 min) and FGT peak (1.1 min). MRM chromatograms of blank HLMs plus FGT (**C**).

**Figure 5 molecules-28-02322-f005:**
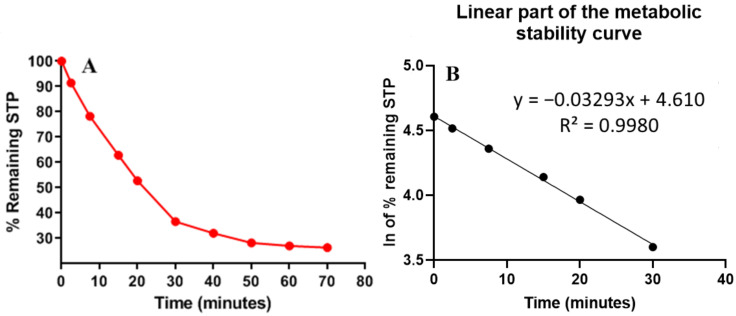
The STP metabolic stability curve in HLMs (**A**) and ln calibration curve showing the regression equation (**B**).

**Figure 6 molecules-28-02322-f006:**
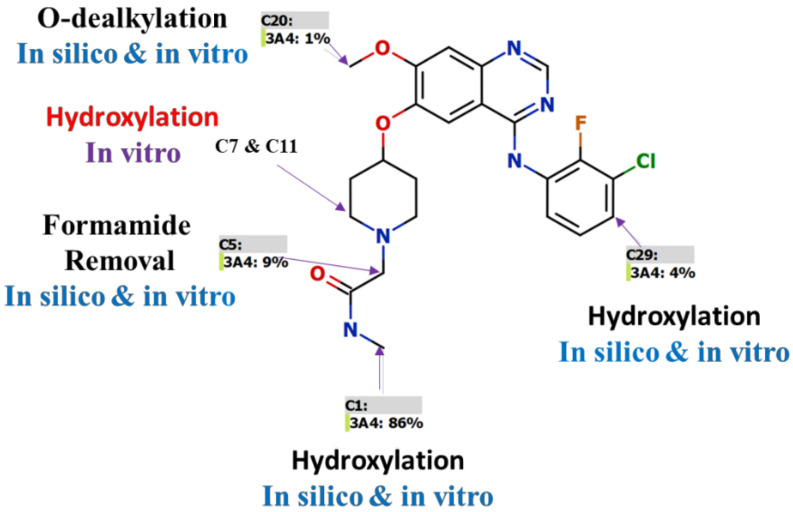
The STP metabolic soft spots that were identified by in silico software and confirmed by in vitro metabolic incubations.

**Table 1 molecules-28-02322-t001:** Different experiments for separation of analyte peaks.

Analytes	ACN	Methanol	Protein Precipitation	Solid Phase Extraction	PFP Column	C18 or C8 Column
SPT	2.17 min	2.81 min	High recovery	Low recovery	2.17 min	3.75 min
	Good	Tailed	Reproducible	Unreproducible	Perfect	Tailed
FGT	1.24 min	2.64 min	High recovery	Low recovery	1.24 min	2.14 min
	Good	Overlapped	Reproducible	Unreproducible	Perfect	Perfect

**Table 2 molecules-28-02322-t002:** Back-calculation data of six repeats (eleven calibration standards and QCs) of SPT.

SPT Nominal Concentrations (ng/mL)	Mean	SD	Precision (RSD %)	Accuracy (RE %)	Recovery
1(LLQC)	1.06	0.02	1.92	5.96	105.96
3(LQC)	3.11	0.12	3.72	3.59	103.59
15	15.37	0.18	1.19	2.45	102.45
50	50.40	1.41	2.80	0.81	100.81
100	152.07	1.56	1.02	1.38	101.38
300	295.03	2.48	0.84	−1.66	98.34
500	502.85	7.92	1.57	0.57	100.57
900 (MQC)	894.72	2.63	0.29	−0.59	99.41
1500	1520.98	19.26	1.27	1.40	101.40
2400 (HQC)	2394.09	17.45	0.73	−0.25	99.75
3000	3027.67	42.18	1.39	0.92	100.92
% Recovery					101.33 ± 2.11

**Table 3 molecules-28-02322-t003:** Precision and accuracy (intraday and interday) results of established method.

STP in HLM Matrix (ng/mL)	Intraday Assay (Twelve Replicates in the Same Day)				Interday Assay (Six Replicates Over Three Consecutive Days)			
	1 (LLQC)	3 (LQC)	900 (MQC)	2400 (HQC)	1 (LLQC)	3 (LQC)	900 (MQC)	2400 (HQC)
Mean	1.06	3.11	894.72	2394.09	1.03	3.22	886.99	2365.33
SD	0.02	0.12	2.63	17.45	0.06	0.20	6.17	17.61
Precision (RSD %)	1.92	3.72	0.29	0.73	6.31	6.29	0.70	0.74
Accuracy (RE %)	5.96	3.59	−0.59	−0.25	2.57	7.28	−1.45	−1.44
Recovery (%)	105.96	103.59	99.41	99.75	102.57	107.28	98.55	98.56

**Table 4 molecules-28-02322-t004:** Stability data of SPT.

Analyte	Concentration (ng/mL)	Freeze–Thaw Stability (3 Cycles, −80 °C)	Short-Term Stability (4 h at Room T)	Long-Term Stability (−80 °C for 28 d)	Autosampler Stability (24 h at 15 °C)
SPT	LQC (3)	98.8 ± 2.4	99.4 ± 2.5	98.6 ± 2.3	99.7 ± 2.7
	HQC (2400)	101.3 ± 2.2	99.2 ± 2.7	100.5 ± 2.8	99.8 ± 2.5

**Table 5 molecules-28-02322-t005:** STP metabolic stability.

Time (min)	Mean ^a^ (ng/mL)	X ^b^	ln X	Linearity Parameters
**0**	584	100.00	**4.61**	Regression equation: y = −0.0329x + 4.6097
**2.5**	534	91.44	**4.52**	
**7.5**	457	78.25	**4.36**	R² = 0.998
**15**	367	62.84	**4.14**	
**20**	308	52.74	**3.97**	Slope: −0.0329
**30**	214	36.64	**3.60**	
40	187	32.02	3.47	t_1/2_: 21.07 min
50	165	28.25	3.34	Cl_int_: 38.48 mL/min/kg
60	158	27.05	3.30	
70	154	26.37	3.27	

Notes:^: a^ Mean of three repeats; ^b^ X: mean of STP percentage remaining of three replicates. Linear range (0–30 min) is indicated by bold font.

**Table 6 molecules-28-02322-t006:** MRM parameters for the determination of SPT and FGT (IS).

Analyte	Rt	Ion Mode	Precursor (*m*/*z*)	Quantification Traces (*m*/*z*)	Qualification Traces (*m*/*z*)	Cone Voltage (V)	Collision Energy (CE, eV)
SPT	2.17	+ve	474.0	320.0	149.9	38	24/28
FGT (IS)	1.24	+ve	426.0	291.0	323.0	38	24/38

## Data Availability

All data are available within the manuscript.

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
