# Peer review of "Development of an LC-MS/MS Method for Quantification of Sapitinib in Human Liver Microsomes: In Silico and In Vitro Metabolic Stability Evaluation"

_molecules, 2023, doi:10.3390/molecules28052322_

Round 1
Reviewer 1 Report
The manuscript submitted by Attwa et al. is devoted to development of a method for quantification of sapitinib in human liver microsomes. The presented study is presented quite well and can be used in labs dealing with anticancer drugs. Nevertheless, there are some points which, in my opinion, need correction and explanation. These are following:
1) The title of the article seems to be not very appropriate, I recommend rephrase it as follows: "... quantification of sapitinib in HLMs: In silico and In vitro Metabolic stability evaluation".
Also, I don't recommend using the abbreviation HLMs in the title. I suppose that it is better to use the full name.
2) There are some studies related to the topic of the submitted manuscript. First of all, this is the study published in 2019 by the same authors where metabolites of sapotinib in HLMs were found and identified (RSC Adv, 2019, 9, 32995 (DOI: 10.1039/c9ra03926k). The second one is the study in Xenobiotica, 2014, v44, p1083 (DOI: 10.3109/00498254.2014.938257).
I think that the novelty of the presented study along with the comparison of the obtained results must be clearly mentioned and discussed.
3) Abstract, line 36: as a consequence of the comment #2 - why authors do state that the presented manuscript is the first study on metabolic stability? It has already been shown that SPT is not stable in HLMs (see RSC Adv., 2019).
4) Page 2, lines 53-54: The font here (Times New Roman) is different from that user for other text.
5) Line 65: the text <Fig.1> should be removed.
6) Fig. 1, title: Please correct "selpercatinib".
7) Line 68: "Extensive reviewing the literature, there are no reported analytical..." - please correct the language. Also, I recommend revision of the language by a native speaker or a special service.
8) Line 69: As for the absence of any LC-MS/MS methods published in literature, indeed, there are no data on this. Nevertheless, in the study published in Xenobiotica, 2014 (see above) there is some information of LC conditions and mass spectrometry detection parameters for SPT. At least, the data published in this article must be discussed with respect to the presented study. Also, I suppose that due to the fact that SPT is a recognized drug, its pharmacokinetics has been studied, but the data may be not published. This is also should be mentioned in the text.
9) Line 71: Repeated SPT abbreviation.
10) Page 3, line 110: An extra symbol | is typed here.
11) Page 3, equation 1: I recommend using subscripts in the formula (ktotal, kw).
12) Page 4, line 140: lower % of ACN?
13) Line 143: There are no data and no discussion on testing the SPT and IS on an ordinary C18 phase column. Meanwhile, this phase was used for separation of SPT in the previously published article (RSC Adv, 2019). Of course, it is the authors who decide which column to be used, but PFP phases are not commonly utilized in analytical laboratories, thus, a rationale should be provided to explain the fluorine-containing sorbent.
14) Section 2.3.1: The flow rate used during the analysis is better to be given here but not in the introduction.
15) Lines 154-155: formula but not formulae.
16) Line 158: If the measurement unit is liters per hour, then, the abbreviation L/h should be used instead of L/H.
17) Lines 167-170: I recommend to provide the MRM parameters specific for each compound and MRM transition as a Table.
18) Page 6, line 186: "HLMs were..."
19) Section 2.6: Please specify here the volume of the calibration sample used for preparation by protein precipitation.
20) Page 8, section 3.1: This section seems to have low value in the presented format. Thus, what can be concluded from the predicted metabolic stability? There are several software examples and online services which can provide tentative molecule sites which can be metabolized or even metabolites. There is no any information confirming the predicted metabolic stability.
In this case, I suggest either to add an extensive discussion of the obtained in silico data with the experimental data provided in RSC Adv, 2019, or to remove it at all followed by correction of the title of the article and other related text.
21) Page 9, Fig. 3: Please provide more information on the figure, namely:
> what are the color sectors in the pie diagram? What do the notifications 3A4, 2D6, etc. mean?
> Please draw lines denoting carbon atoms of the molecule more more distinctly. Also, I recommend to make the full numeration or to provide an additional figure, for example, in Supplementary where all atoms of the molecule would be numbered.
22) Section 3.2: The volume of a column having the size of 4.6x150 mm is c.a. 2.5 mL. Given the flow rate 0.4 mL/min, how is it possible that the RTs of the analytes can be 1.24 min and 2.17 min?
23) Page 11, line 329: An extra bracket is here (R2): 0.9993).
24) Table 1 and 2: In my opinion, Accuracy is the parameter calculated as (Found/Nominal)*100%. Thus, 1.06 ng/mL found for LLOQ = 1 ng/mL would mean Accuracy = 106%. The presented value (and others) of 5.96% means the bias which should be either mentioned in the column heading or recalculated.
25) Page 12: Please remove the abbreviation "conc." here and further in the text and use the full word.
26) Line 365: Please correct LN into ln here and further in the text and in Fig. 5.
27) The number indicating the Equation 8 is absent.
28) Page 13, line 388: I suppose that the statement that the presented study is the first analytical method for SPT quantification is arguable. As I wrote it above, the pharmacokinetics of the drug has been studied, and there are publication on this. Unfortunately, these publications mainly deal with activity of the drug, or pharmacokinetics is provided without details on the analysis (e.g. Tjulandin S, Moiseyenko V, Semiglazov V, et al. Phase I, dose-finding study of AZD8931, an inhibitor of EGFR (erbB1), HER2 (erbB2) and HER3 (erbB3) signaling, in patients with advanced solid tumors. Invest New Drugs. 2014;32(1):145-153. doi:10.1007/s10637-013-9963-6). Nevertheless, I recommend authors to avoid such a strong statement and correct the concluding part of their manuscript.
As a summary, I think that the major drawback of the presented study is the absence of the discussion on the metabolism of SPT which has been published by the same authors earlier. Also, I suppose that the HPLC conditions should be clearly described.
Author Response
Manuscript ID: molecules- 2248281
Authors’ response
We thank the editor for this opportunity to improve our manuscript and be considered again for publication in the Molecules Journal. We give the below-detailed answers to each question raised by reviewer # 1. All replies to the comments were highlighted in yellow color in the revised manuscript.
Reviewer # 1
Comments and Suggestions for Authors
The manuscript submitted by Attwa et al. is devoted to the development of a method for the quantification of sapitinib in human liver microsomes. The presented study is presented quite well and can be used in labs dealing with anticancer drugs. Nevertheless, there are some points which, in my opinion, need correction and explanation. These are following:
Authors’ response
We appreciate the reviewer’s words and his/her suggestions to improve our manuscript. We give below our answer to his/her concerns.
Point # 1
1) The title of the article seems to be not very appropriate, I recommend rephrase it as follows: "... quantification of sapitinib in HLMs: In silico and In vitro Metabolic stability evaluation".
Also, I don't recommend using the abbreviation HLMs in the title. I suppose that it is better to use the full name.
Authors’ response
The title was changed as requested to
Development of an LC–MS/MS method for Quantification of Sapitinib in human liver microsomes: In silico and In vitro Metabolic Stability Evaluation
Point # 2
2) There are some studies related to the topic of the submitted manuscript. First of all, this is the study published in 2019 by the same authors where metabolites of sapotinib in HLMs were found and identified (RSC Adv, 2019, 9, 32995 (DOI: 10.1039/c9ra03926k). The second one is the study in Xenobiotica, 2014, v44, p1083 (DOI: 10.3109/00498254.2014.938257).
I think that the novelty of the presented study along with the comparison of the obtained results must be clearly mentioned and discussed.
Authors’ response
- We added a discussion section to explain the importance and novelty of the current work and we discussed the results of the previous qualitative work with the results of the current manuscript.
A fully validated LC-MS/MS analytical method was established for estimation of SPT in HLMs matrix in the range from 1 ng/mL to 3000 ng/mL. It can be used for TDM of SPT as the maximum concentration of SPT in human plasma is 3000 ng/mL. The P450 StarDrop software results of CSL (0.9947) revealed high SPT metabolic lability that matched with the metabolic stability experiment (Clint: 38.48 mL/min/kg) that is considered higher than previously studied tyrosine kinase inhibitors (ex Tandutinib)[33-36]. The in silico results showed that N-methylacetamide group (C1 and C5) is the main reason of SPT metabolic lability that was previously approved in the metabolites identification study [11]. In our previous work about SPT metabolites identification, six metabolites were identified in the same metabolic spots that were predicted by P450 StarDrop software. The metabolic hydroxylation of C7 and C11 was seen in practical experiments (Figure 6).
Figure 6. The STP metabolic soft spots that were identified by in silico software and confirmed by in vitro metabolic incubations.
From all the results of the previous qualitative work for SPT metabolites identification and the current work for quantitative estimation of the rate of SPT metabolism, in silico software (StarDrop software) could be used in an efficient way to guide the practical experiments to save time and resources especially in the first steps for drug design steps.
- We rephrased many parts of the manuscript to clarify the target and to verify the results and to link it and compare the current work with the literature.
- We added two paragraphs to the introduction to report the two articles explaining the difference between the current work for metabolic stability and the previous work for metabolites and reactive metabolites of sapitinib:
Metabolic stability of a drug is the vulnerability to metabolism and is stated as in vitro half-life (t1/2) and intrinsic clearance (CLint). CLint is defined as the liver ability to metabolize the drug in the blood. Half-life (t1/2) is defined as the time needed for 50% metabolism of the parent drug. The in vitro Clint and t1/2 in HLMs were computed by an ‘in vitro t1/2’ approach using the ‘well-stirred’ model [5, 6] as it is the frequently used model in drug metabolism experiments owing to its simplicity. These parameters (Clint and in vitro t1/2) could be used for computing various physiological parameters (e.g., in vivo t1/2 and liver clearance). The drug bioavailability evaluation provides a good approach to proposing it’s in vivo metabolic reactions. If the examined drug shows fast metabolic rate, it will exhibit low in vivo bioavailability value and a short duration of action [7-10].
Literature review revealed that there is no reported article for a fully validated LC-MS/MS method for SPT quantification in HLMs matrix. There are two articles published for qualitative identification of metabolites of SPT [11, 12]. Peter et al, reported the LC-MS/MS for quantitative estimation of SPT and O-desmethyl metabolite without providing any details about the analytical chromatographic parameters and no method validation was performed [12]. Attwa et al, reported qualitative analytical method for profiling of metabolites and reactive intermediates of SPT without determining the rate of metabolism with time [11]. Also, high concentration of SPT (10 µM/mL) was used in metabolic profiling studies which is not performed for metabolic stability studies of SPT ((1 µM/mL) to be lower than the Michaelis–Menten constant to construct a linear relationship between the STP metabolic rate and the time of metabolic incubation.
Point # 3
3) Abstract, line 36: as a consequence of the comment #2 - why authors do state that the presented manuscript is the first study on metabolic stability? It has already been shown that SPT is not stable in HLMs (see RSC Adv., 2019).
Authors’ response
- We rephrased the sentence to be more accurate as the following:
Literature review demonstrated that the developed analytical method is the first developed LC-MS/MS method for quantification of SPT in HLMs matrix with the application to SPT metabolic stability evaluation.
- We added a discussion section to explain the importance of the current work and the great difference of the qualitative work of the previous article and the quantitative work of the current manuscript.
- We rephrased many parts of the manuscript to clarify the target and to verify the results and to link it and compare the current work with the literature.
- The RSC Adv. article was a qualitative work for identification of metabolites and reactive intermediates whatever the conc. In the previous study we used 10 µM of sapitinib conc. in incubation but the current study for metabolic stability is different from metabolites identification that 1 µM of sapitinib was used.
- We did not develop an LC-MS/MS for quantitative estimation of SPT to measure the metabolism rate.
- In the current work a fully validated LC-MS/MS was developed to measure the rate of SPT degradation with HLMs to calculate the metabolic stability.
- We added a paragraph to the introduction explain the importance of metabolic stability estimation:
Metabolic stability of a drug is the vulnerability to metabolism and is stated as in vitro half-life (t1/2) and intrinsic clearance (CLint). CLint is defined as the liver ability to metabolize the drug in the blood. Half-life (t1/2) is defined as the time needed for 50% metabolism of the parent drug. The in vitro Clint and t1/2 in HLMs were computed by an ‘in vitro t1/2’ approach using the ‘well-stirred’ model [5, 6] as it is the frequently used model in drug metabolism experiments owing to its simplicity. These parameters (Clint and in vitro t1/2) could be used for computing various physiological parameters (e.g., in vivo t1/2 and liver clearance). The drug bioavailability evaluation provides a good approach to proposing it’s in vivo metabolic reactions. If the examined drug shows fast metabolic rate, it will exhibit low in vivo bioavailability value and a short duration of action [7-10].
Point # 4
4) Page 2, lines 53-54: The font here (Times New Roman) is different from that user for other text.
Authors’ response
It was corrected as requested.
Point # 5
5) Line 65: the text <Fig.1> should be removed.
Authors’ response
It is removed as requested.
Point # 6
6) Fig. 1, title: Please correct "selpercatinib".
Authors’ response
Sorry for the typo mistake. It is corrected as requested.
Point # 7
7) Line 68: "Extensive reviewing the literature, there are no reported analytical..." - please correct the language. Also, I recommend revision of the language by a native speaker or a special service.
Authors’ response
We corrected the sentence. We revised the last version of the manuscript by a native English speaker in researcher support unit at King Saud University.
Point # 8
8) Line 69: As for the absence of any LC-MS/MS methods published in literature, indeed, there are no data on this. Nevertheless, in the study published in Xenobiotica, 2014 (see above) there is some information of LC conditions and mass spectrometry detection parameters for SPT. At least, the data published in this article must be discussed with respect to the presented study. Also, I suppose that due to the fact that SPT is a recognized drug, its pharmacokinetics has been studied, but the data may be not published. This is also should be mentioned in the text.
Authors’ response
We rephrased the sentence to be more accurate.
We added two paragraphs to the introduction to report the two articles explaining the difference between the current work for metabolic stability and the previous work for metabolites and reactive metabolites of sapitinib:
Literature review revealed that there is no reported article for a fully validated LC-MS/MS method for SPT quantification in HLMs matrix. There are two articles published for qualitative identification of metabolites of SPT [11, 12]. Peter et al, reported the LC-MS/MS for quantitative estimation of SPT and O-desmethyl metabolite without providing any details about the analytical chromatographic parameters and no method validation was performed [12]. Attwa et al, reported qualitative analytical method for profiling of metabolites and reactive intermediates of SPT without determining the rate of metabolism with time [11]. Also, high concentration of SPT (10 µM/mL) was used in metabolic profiling studies which is not performed for metabolic stability studies of SPT ((1 µM/mL) to be lower than the Michaelis–Menten constant to construct a linear relationship between the STP metabolic rate and the time of metabolic incubation.
Point # 9
9) Line 71: Repeated SPT abbreviation.
Authors’ response
We rephrased the whole paragraph and we updated the revised version of the manuscript.
Point # 10
10) Page 3, line 110: An extra symbol | is typed here.
Authors’ response
The typo mistake was corrected; the symbol was removed.
Point # 11
11) Page 3, equation 1: I recommend using subscripts in the formula (ktotal, kw).
Authors’ response
The equation was corrected as requested.
Point # 12
12) Page 4, line 140: lower % of ACN?
Authors’ response
It was corrected as the following:
The organic part (ACN) represented 55% of the mobile phase. Increasing the % of ACN (more than 55%) generated overlapped and poor separated chromatographic peaks, while decreasing the % of ACN (less than 55%) generated long elution time.
Point # 13
13) Line 143: There are no data and no discussion on testing the SPT and IS on an ordinary C18 phase column. Meanwhile, this phase was used for separation of SPT in the previously published article (RSC Adv, 2019). Of course, it is the authors who decide which column to be used, but PFP phases are not commonly utilized in analytical laboratories, thus, a rationale should be provided to explain the fluorine-containing sorbent.
Authors’ response
In the previous study (RSC Advances, 2019), we used gradient method with long elution time to separate closely related metabolites. In the current work (quantitative work), there are some important chromatographic parameters such as peak shape and retention to achieve optimum separation in a short running time. So, the best results were attained using PFP column regarding peak shape and retention time. Also it was stated that Luna PFP(2) offers powerful selectivity not common with traditional alkyl phases (C18, C8). This new Luna chemistry incorporates a pentafluorophenylpropyl ligand bonded to Luna silica. Luna PFP(2) uses several selectivity mechanisms (hydrogen bonding, dipole-dipole, and aromatic) in addition to hydrophobic interactions to greatly increase resolution of difficult compounds, including:halogenated compounds as seen in the following link:
https://www.selectscience.net/product-news/explore-luna-pfp(2)-hplc-columns,-high-performance-silica-based-columns/?artID=13775
The following paragraph was updated in the revised version of the manuscript:
Reversed phase C18 column was tested for separation in the current work (quantitative work), analytes were retained but generated chromatographic peak tailing, poor separation and longer retention time. So, the best results were attained using PFP column regarding peak shape and retention time.
Point # 14
14) Section 2.3.1: The flow rate used during the analysis is better to be given here but not in the introduction.
Authors’ response
The following sentences were transferred from the introduction to section 2.3.1
In the established method, isocratic mobile phase in 3 min run time using less flow rate (0.4 mL) was used. Moreover, the calibration curve was linear in the range of 1 to 3000 ng/mL.
Point # 15
15) Lines 154-155: formula but not formulae.
Authors’ response
The typo mistake was corrected as requested.
Point # 16
16) Line 158: If the measurement unit is liters per hour, then, the abbreviation L/h should be used instead of L/H.
Authors’ response
The measurement unit is liters per hour, then, the abbreviation was corrected to L/h in the revised version as requested.
Point # 17
17) Lines 167-170: I recommend to provide the MRM parameters specific for each compound and MRM transition as a Table.
Authors’ response
The following sentence and table were updated in the revised version of the manuscript:
All MRM parameters and mass transition of SPT and FGT (IS) are listed in table 1.
Table 1. MRM parameters for the determination of SPT and FGT (IS).
|
Collision energy (CE, eV)
|
Cone Voltage (V) |
Qualification traces (m/z) |
Quantification traces (m/z) |
Precursor (m/z) |
Ion mode |
Rt |
Analyte |
|
24/28 |
38 |
149.9 |
320.0 |
474.0 |
+ve |
2.17 |
SPT |
|
24/38 |
38 |
323.0 |
291.0 |
426.0 |
+ve |
1.24 |
FGT (IS) |
Point # 18
18) Page 6, line 186: "HLMs were..."
Authors’ response
The typo mistake was corrected as requested.
Point # 19
19) Section 2.6: Please specify here the volume of the calibration sample used for preparation by protein precipitation.
Authors’ response
The volume of the calibration samples was 1 mL to simulate the incubation volume and 2 mL of ACN was added to all samples.
The following sentence was updated in the revised version of the manuscript:
First: two mL of ACN was added to 1 mL of the STP calibration levels, quality controls or unknown samples.
Point # 20
20) Page 8, section 3.1: This section seems to have low value in the presented format. Thus, what can be concluded from the predicted metabolic stability? There are several software examples and online services which can provide tentative molecule sites which can be metabolized or even metabolites. There is no any information confirming the predicted metabolic stability.
In this case, I suggest either to add an extensive discussion of the obtained in silico data with the experimental data provided in RSC Adv, 2019, or to remove it at all followed by correction of the title of the article and other related text.
Authors’ response
We updated many parts of the manuscript to clarify the role of the in silico software.
We added the following discussion as requested:
Discussion
A fully validated LC-MS/MS analytical method was established for estimation of SPT in HLMs matrix in the range from 1 ng/mL to 3000 ng/mL. It can be used for TDM of SPT as the maximum concentration of SPT in human plasma is 3000 ng/mL. The P450 StarDrop software results of CSL (0.9947) revealed high SPT metabolic lability that matched with the metabolic stability experiment (Clint: 38.48 mL/min/kg) that is considered higher than previously studied tyrosine kinase inhibitors (ex Tandutinib)[33-36]. The in silico results showed that N-methylacetamide group (C1 and C5) is the main reason of SPT metabolic lability that was previously approved in the metabolites identification study [11]. In our previous work about SPT metabolites identification, six metabolites were identified in the same metabolic spots that were predicted by P450 StarDrop software. The metabolic hydroxylation of C7 and C11 was seen in practical experiments (Figure 6).
Figure 6. The STP metabolic soft spots that were identified by in silico software and confirmed by in vitro metabolic incubations.
From all the results of the previous qualitative work for SPT metabolites identification and the current work for quantitative estimation of the rate of SPT metabolism, in silico software (StarDrop software) could be used in an efficient way to guide the practical experiments to save time and resources especially in the first steps for drug design steps.
Point # 21
21) Page 9, Fig. 3: Please provide more information on the figure, namely:
> what are the color sectors in the pie diagram? What do the notifications 3A4, 2D6, etc. mean?
> Please draw lines denoting carbon atoms of the molecule more distinctly. Also, I recommend to make the full numeration or to provide an additional figure, for example, in Supplementary where all atoms of the molecule would be numbered.
Authors’ response
The following figure and details were updated in the revised manuscript:
WhichP450 model predicts the major metabolizing isoform (CYP3A4) for SPT metabolism as approved by the pie chart (Figure 3 A). Regioselectivity map indicates predicted sites of metabolism for SPT (Figure 3 B). The metabolic landscape (Figure 3C) predicts the SPT metabolic lability of the active sites to improve the understanding of the SPT metabolic rate.
Figure 3. WhichP450 model predicts the major metabolizing isoform (CYP3A4) for SPT metabolism as approved by the pie chart (A). Regioselectivity map indicates predicted sites of metabolism for SPT (B). Metabolic landscape showing CSL of SPT (0.9797) revealed the high metabolic rate (C). Chemical structure of SPT with numbering system (D). These data were performed using StarDrop P450 metabolism model.
Also the discussion part were added to clarify the main target of the current manuscript.
Point # 22
22) Section 3.2: The volume of a column having the size of 4.6x150 mm is c.a. 2.5 mL. Given the flow rate 0.4 mL/min, how is it possible that the RTs of the analytes can be 1.24 min and 2.17 min?
Authors’ response
The column is packed with stationary phase particles (3µm particle size), so the void volume is very small. At the same time, we used a high percent of organic solvent (55% ACN), so it fastens the moving through the column. By controlling pH and organic solvent, we can control the retention time.
The following figure is application from the company website:
Luna PFP using 0.2 mL flow rate and the Rt is less than 2 min.
Point # 23
23) Page 11, line 329: An extra bracket is here (R2): 0.9993).
Authors’ response
The typo mistake was corrected as requested.
Point # 24
24) Table 1 and 2: In my opinion, Accuracy is the parameter calculated as (Found/Nominal)*100%. Thus, 1.06 ng/mL found for LLOQ = 1 ng/mL would mean Accuracy = 106%. The presented value (and others) of 5.96% means the bias which should be either mentioned in the column heading or recalculated.
Authors’ response
Thank you for your comment. We mentioned this in the column heading as requested.
Accuracy and precision of the current UPLC-TQD MS method were expressed as % error and % relative standard deviation (% RSD), respectively as calculated in equation 4 and equation 5, respectively.
Equation (4)
Equation (5)
So we changed the column headings in table 1 and table 2.
Point # 25
25) Page 12: Please remove the abbreviation "conc." here and further in the text and use the full word.
Authors’ response
This was done all over the manuscript as requested.
Point # 26
26) Line 365: Please correct LN into ln here and further in the text and in Fig. 5.
Authors’ response
This was done all over the manuscript and figure 5 as requested.
Point # 27
27) The number indicating the Equation 8 is absent.
Authors’ response
This was updated in the revised manuscript:
as was calculated in equation 8.
Equation (8)
Point # 28
28) Page 13, line 388: I suppose that the statement that the presented study is the first analytical method for SPT quantification is arguable. As I wrote it above, the pharmacokinetics of the drug has been studied, and there are publication on this. Unfortunately, these publications mainly deal with activity of the drug, or pharmacokinetics is provided without details on the analysis (e.g. Tjulandin S, Moiseyenko V, Semiglazov V, et al. Phase I, dose-finding study of AZD8931, an inhibitor of EGFR (erbB1), HER2 (erbB2) and HER3 (erbB3) signaling, in patients with advanced solid tumors. Invest New Drugs. 2014;32(1):145-153. doi:10.1007/s10637-013-9963-6). Nevertheless, I recommend authors to avoid such a strong statement and correct the concluding part of their manuscript.
Authors’ response
We rephrased many parts of the manuscript
We added the following paragraph to the introduction:
Literature review revealed that there is no reported article for a fully validated LC-MS/MS method for SPT quantification in HLMs matrix. There are two articles published for qualitative identification of metabolites of SPT [11, 12]. Peter et al, reported the LC-MS/MS for quantitative estimation of SPT and O-desmethyl metabolite without providing any details about the analytical chromatographic parameters and no method validation was performed [12]. Attwa et al, reported qualitative analytical method for profiling of metabolites and reactive intermediates of SPT without determining the rate of metabolism with time [11]. Also, high concentration of SPT (10 µM/mL) was used in metabolic profiling studies which is not performed for metabolic stability studies of SPT ((1 µM/mL) to be lower than the Michaelis–Menten constant to construct a linear relationship between the STP metabolic rate and the time of metabolic incubation.
Point # 29
As a summary, I think that the major drawback of the presented study is the absence of the discussion on the metabolism of SPT which has been published by the same authors earlier. Also, I suppose that the HPLC conditions should be clearly described.
Authors’ response
We updated many parts of the manuscript to clarify the importance of the current work and to approve that it different and completes the previous work.
We added the following discussion as requested:
Discussion
A fully validated LC-MS/MS analytical method was established for estimation of SPT in HLMs matrix in the range from 1 ng/mL to 3000 ng/mL. It can be used for TDM of SPT as the maximum concentration of SPT in human plasma is 3000 ng/mL. The P450 StarDrop software results of CSL (0.9947) revealed high SPT metabolic lability that matched with the metabolic stability experiment (Clint: 38.48 mL/min/kg) that is considered higher than previously studied tyrosine kinase inhibitors (ex Tandutinib) [33-36]. The in silico results showed that N-methylacetamide group (C1 and C5) is the main reason of SPT metabolic lability that was previously approved in the metabolites identification study [11]. In our previous work about SPT metabolites identification, six metabolites were identified in the same metabolic spots that were predicted by P450 StarDrop software. The metabolic hydroxylation of C7 and C11 was seen in practical experiments (Figure 6).
Figure 6. The STP metabolic soft spots that were identified by in silico software and confirmed by in vitro metabolic incubations.
From all the results of the previous qualitative work for SPT metabolites identification and the current work for quantitative estimation of the rate of SPT metabolism, in silico software (StarDrop software) could be used in an efficient way to guide the practical experiments to save time and resources especially in the first steps for drug design steps.
Kind regards
Dr. Mohamed Attwa
Dept. Pharm. Chem., College of Pharmacy, King Saud University,
P.O. Box 2457,
Riyadh, 11451, Saudi Arabia
mzeidan@ksu.edu.sa

Reviewer 2 Report
Please find the attachemnt for my comments,

Author Response
Manuscript ID: molecules- 2248281
Authors’ response
We thank the editor this opportunity to improve our manuscript and be considered again for publication in Molecules Journal. We give below detailed answers to each question raised by reviewer # 2. All replies to the comments were highlighted in yellow color in the revised manuscript.
Reviewer # 2
Comments and Suggestions for Authors
The manuscript entitled "Development of LC–MS/MS method for Quantification of Sapitinib: In silico and In vitro Metabolic Stability Evaluation in HLMs" deals with method development and validation by LC-MS/MS for sapitinib understanding its metabolic stability evaluation in HLMs. It is well written and organized, however it needs some revision before it can be considered for publication:
Authors’ response
We appreciate the reviewer’s words and his/her suggestions to improve our manuscript. We give below our answer to his/her concerns.
We rearranged the whole manuscript adding more information in each section. We clarified many vague information. We added the requested data.
Point # 1
- The validation of the method has been fully covered by the authors: How many experiments, in the optimizing procedure, were conducted e.g. for each solvent, extraction time, sample volume? and data including the above should be given, as a Table.
Authors’ response
Different experiments were performed to select the most optimum condition for separation, extraction and analysis of SPT and FGT chromatographic peaks in a perfect shape and in a short running time as mentioned in table 1. All other experiments are mentioned in details in the validation steps of the established analytical methodology.
Point # 2
- What is the rationale for selecting normal phase column (polar HILIC column) during method development process? It looks better to remove the statement.
Authors’ response
The sentence was removed as requested as these analytes are expected that it will not be retained.
Point # 3
- What is the reason for selecting too many (eleven points) calibration standards?
Authors’ response
We construct the calibration curve using eight points and we used different four points as quality controls. We generated the regression equation using the eight points as a calibrations standards then we back calculated all data points including the quality controls as unknowns. The calibration curve constructed of only eight points without the quality controls (LQC, MQC and HQC). We update the data and the sentences to remove the misunderstand.
STP calibration standards were prepared by multistep dilution of STP WK2 and STP WK3 with the deactivated HLMs matrix generating eight levels: 1, 15, 50, 150, 300, 500, 1500 and 3000 ng/mL keeping the HLMs matrix volume not less than 90% of the prepared solution in order to decrease the influence of dilution if compared to real samples.
Point # 4
- As per the regulatory guidelines, quality control sample concentrations must be different than the calibration curve concentrations. In this study, QC samples LLQC (1 ng/mL), LQC (3 ng/mL), MQC (900 ng/mL), HQC (2400 ng/mL) are included in the calibration points. Justify it.
Authors’ response
The calibration curve constructed of only eight points without the quality controls (LQC, MQC and HQC).
We update the data and the sentences to remove the misunderstand.
STP calibration standards were prepared by multistep dilution of STP WK2 and STP WK3 with the deactivated HLMs matrix generating eight levels: 1, 15, 50, 150, 300, 500, 1500 and 3000 ng/mL keeping the HLMs matrix volume not less than 90% of the prepared solution in order to decrease the influence of dilution if compared to real samples. These STP calibration standards were used for making a calibration curve. Four STP levels were selected as quality controls (QCs) for the STP validation procedure: LLQC (1 ng/mL), LQC (3 ng/mL), MQC (900 ng/mL) and HQC (2400 ng/mL). QCs were used as unknowns and the concentration were determined using a freshly prepared STP calibration curve. One hundred microliters of FGT WK1 was added as the IS (1000 ng/mL) to 1 mL of calibration levels and QCs.
The sensitivity and linearity of the established UPLC-TQD MS method were evaluated by injecting freshly prepared calibration curves (eight standards) and quality controls of STP in HLMs matrix in the same day then back calculating of all data unknowns using the linear regression equation of each curve.
Point # 5
- Did the authors check LOD and LOQ by signal to noise ratio method as well?
Authors’ response
Yes, we tried and we found results are not rational. Using MRM analyzer mode for detection and analysis of analytes ions make the system almost without noise that gave a false indication for the LOD and LOQ. In most of the cases, we depended on the practical estimation of LOD and LOQ that confirmed linearity and the validation of the analytical method.
In the following figure: the S/N ratio at LQC (3 ng/mL) is 166.13, so the LOQ should be 0.18 ng/mL that is not achieved practically due to the large value of SD and RSD due to the wide range of the calibration curve that depends mainly in peak are ratio.
Figure: S/N ratio at LQC (3 ng/mL) is 166.13.
Point # 6
- What is the reason for selecting 1/x as a weighing factor on calibration curve?
Authors’ response
Due to the wider range for calibration curve from 1 to 3000 ng of STP. So we tried to analyze data without weighing, there was a large deviation (not accepted) for smaller concentrations (LLQC and LQC). So, we analyzed using 1/x weighing factor and all validation parameters were optimized.
Point # 7
- Authors need to include the method stability parameter in the validation exercise as per the USFDA regulatory guidelines.
Authors’ response
We expected that request. We had all data before submitting the manuscript except one-month stability for stock solution that was performed during revision stage. The stability data was updated in the revised manuscript.
Point # 8
- Include some of the relevant recent references such as 10.3390/molecules25215004, 10.1039/C9RA03926K, 10.1039/C9RA08121F, 10.3109/00498254.2014.938257 etc.
Authors’ response
All proposed articles were cited in the revised version of the manuscript as requested.
Point # 9
- In general, it is well written, but there are some grammars and typing/spacing errors, a few of them are at line no 56 double bracket for SPT, at line no 110 symbol must be removed before Vacuum, in Table 3, notation for b is not mentioned but it was mentioned in the noted under the table, these should be corrected. The English is generally satisfactory but a native speaker should read the paper and correct some sentences.
Authors’ response
All the requested changes were performed as requested. The whole manuscript was revised for grammatical and typo errors. We revised the last version of the manuscript by a native English speaker in researcher support unit at King Saud University.
Point # 10
- This work is interesting with novel objective and the authors have done a number of experiments to establish their hypotheses. Still the article has some queries needs to be addressed before publishing.
Authors’ response
Thank you very much for your time and effort helping us to improve our manuscript. Hoping our replies and changes in the manuscript would be enough for the manuscript to be published. We added many information to the revised version of the manuscript.
Kind regards
Dr. Mohamed Attwa
Dept. Pharm. Chem., College of Pharmacy, King Saud University,
P.O. Box 2457,
Riyadh, 11451, Saudi Arabia
mzeidan@ksu.edu.sa

Round 2
Reviewer 1 Report
The revised manuscript contains all corrections made in accordance with my comments. After considering the revision, I'd like to make several recommendations which, in my opinion, would improve the overall quality of the article.
1) Page 4, lines 163-173: The text presented here (starting from the sequence "Reversed phase C18 column ..." as well as table 1 is most suitable for the section "Results and discussion". I just remind you that the part "Materials and Methods" contains the final conditions used in the study. So I recommend just to provide here a description of the final LC conditions and move the mentioned piece of text and Table 1 further.
2) Similar comment is to the section 2.3.2: Just leave the facts on the detection of the compounds by mass spectrometer. I suppose that Fig. 2 is better to be moved to the "Results and Discussion".
3) Page 14, section 4, lines 432-433: A method developed for quantification of SPT in a specific matrix (e.g. HLM) cannot be used for quantification of the drug in another biologivcal matrix (plasma). Thus, I suppose this statement to be wrong and needing corrections.
And my final concern is that the presented manuscript is a good copy of a study published by the same authors in the same Special Issue (Molecules 2023, 28(4), 1641). The structure of the manuscript is the same, so, I recommend the Editors to run the submission through the antiplagiarism service.
If neglect the latter comment, I suppose that the submission is suitable for publication.
Author Response
Molecules Journal
Manuscript ID: molecules- 2248281
Authors’ response
We thank the editor this opportunity to improve our manuscript and be considered again for publication in Molecules Journal. We give below detailed answers to each question raised by reviewer # 1. All replies to the comments were highlighted in yellow color in the revised manuscript.
Reviewer # 1
Comments and Suggestions for Authors
The revised manuscript contains all corrections made in accordance with my comments. After considering the revision, I'd like to make several recommendations which, in my opinion, would improve the overall quality of the article.
Authors’ response
We appreciate the reviewer’s words and his/her suggestions to improve our manuscript. We give our answer to his/her concerns below.
Point # 1
- Page 4, lines 163-173: The text presented here (starting from the sequence "Reversed phase C18 column ..." as well as table 1 is most suitable for the section "Results and discussion". I just remind you that the part "Materials and Methods" contains the final conditions used in the study. So I recommend just to provide here a description of the final LC conditions and move the mentioned piece of text and Table 1 further.
The changes were performed as requested.
Point # 2
- Similar comment is to section 2.3.2: Just leave the facts on the detection of the compounds by mass spectrometer. I suppose that Fig. 2 is better to be moved to the "Results and Discussion".
The changes were performed as requested.
Point # 3
- Page 14, section 4, lines 432-433: A method developed for quantification of SPT in a specific matrix (e.g. HLMs) cannot be used for quantification of the drug in another biological matrix (plasma). Thus, I suppose this statement to be wrong and needing corrections.
The changes were performed as requested. The sentence was removed as requested.
And my final concern is that the presented manuscript is a good copy of a study published by the same authors in the same Special Issue (Molecules 2023, 28(4), 1641). The structure of the manuscript is the same, so, I recommend the Editors to run the submission through the antiplagiarism service.
If neglect the latter comment, I suppose that the submission is suitable for publication.
The similarity percent was checked using the Plagscan program as seen in the attached file and the percent of similarity was 20.1%.
Kind regards
Dr. Mohamed Attwa
Dept. Pharm. Chem., College of Pharmacy, King Saud University,
P.O. Box 2457, Riyadh, 11451, Saudi Arabia
mzeidan@ksu.edu.sa

Reviewer 2 Report
The authors have addressed all of the comments raised, and the manuscript can be accepted in its current form.
Author Response
Molecules Journal
Manuscript ID: molecules- 2248281
Authors’ response
We thank the editor this opportunity to improve our manuscript and be considered again for publication in Molecules Journal. We thank the reviewer # 2 for acceptance for our manuscript and for improving it to the current format.
Reviewer # 2
Comments and Suggestions for Authors
The authors have addressed all of the comments raised, and the manuscript can be accepted in its current form.
Authors’ response
Thank you very much for your time and effort helping us to improve our manuscript.
Kind regards
Dr. Mohamed Attwa
Dept. Pharm. Chem., College of Pharmacy, King Saud University,
P.O. Box 2457,
Riyadh, 11451, Saudi Arabia
mzeidan@ksu.edu.sa
